# Increase in Mutual Information During Interaction with the Environment Contributes to Perception

**DOI:** 10.3390/e21040365

**Published:** 2019-04-04

**Authors:** Daya Shankar Gupta, Andreas Bahmer

**Affiliations:** 1Biology Department, Camden County College, Blackwood, NJ 08012, USA; 2Comprehensive Hearing Center, ENT Clinic, University of Wuerzburg, 97080 Wuerzburg, Germany

**Keywords:** affordance, surprisal, temporal coupling, sparse coding, Shannon information, temporal processing, embedded cognitive theory

## Abstract

Perception and motor interaction with physical surroundings can be analyzed by the changes in probability laws governing two possible outcomes of neuronal activity, namely the presence or absence of spikes (binary states). Perception and motor interaction with the physical environment are partly accounted for by a reduction in entropy within the probability distributions of binary states of neurons in distributed neural circuits, given the knowledge about the characteristics of stimuli in physical surroundings. This reduction in the total entropy of multiple pairs of circuits in networks, by an amount equal to the increase of mutual information, occurs as sensory information is processed successively from lower to higher cortical areas or between different areas at the same hierarchical level, but belonging to different networks. The increase in mutual information is partly accounted for by temporal coupling as well as synaptic connections as proposed by Bahmer and Gupta (Front. Neurosci. 2018). We propose that robust increases in mutual information, measuring the association between the characteristics of sensory inputs’ and neural circuits’ connectivity patterns, are partly responsible for perception and successful motor interactions with physical surroundings. The increase in mutual information, given the knowledge about environmental sensory stimuli and the type of motor response produced, is responsible for the coupling between action and perception. In addition, the processing of sensory inputs within neural circuits, with no prior knowledge of the occurrence of a sensory stimulus, increases Shannon information. Consequently, the increase in surprise serves to increase the evidence of the sensory model of physical surroundings

## 1. Information Theoretic Basis of Perception and Action

Theories of perception can be broadly classified into two major traditions: One group of theories postulates that preexisting mental states that represent the physical world are essential for perception while the other group posits that a preexisting representation of the physical world is not essential for perception [1]. The latter group, embedded in cognitive theories, which postulates that cognitive processes are deeply rooted in the body’s interactions with the environment [2], is strongly influenced by the ideas of James Gibson [1,2,3,4,5]. 

According to Gibson’s ecological psychology, there are three main guiding principles to understand perception, which are: (i) Perception is direct; (ii) perception is for action; and (iii) perception is of affordances, which are defined as directly perceivable objective features of the environment that provide opportunities for behavior [1,5,6]. Furthermore, according to Gibson’s views, perception does not depend on an internal representation of the external physical surroundings. Based on the principles of ecological psychology, we argue that the organism and environment form a single dynamical system, coupling perception with adaptive actions. In this dynamical system, non-linear interactions between feedforward and feedback flow of information within the nervous system would result in instantaneous representations of interactions with the environment. Such forms of instantaneous representations, in radical embedded cognitive theory, are referred to as action-oriented representations [1]. Interactions between an organism and the environment would lead to sets of hidden states, stored in cortical activity, coding instantaneous representations of interactions. The preceding view is consistent with a non-linear dynamic model of perception, recently proposed by Singer and Lazar [7], according to which spontaneous synaptic activity in the neocortex stores priors, performs high-dimensional processing with non-linear dynamics, and provides efficient, flexible computational strategies. High-dimensional states in the cortex, in the presence of given sensory inputs, collapse into specific low-dimensional substates corresponding to a specific perceptual experience [7]. While the work by Singer and Lazar [8] provides a model for the complex dynamics between the organism and its environment, it is also an important link in understanding current model-driven research in the basis of perception. Hermann von Helmholtz (1866) famously suggested that perception results from a probabilistic inferential process, producing the best fit for a perceptual object based on sensory data [8], which has inspired modern predictive coding theory [9,10,11,12,13,14]. In the predictive coding scheme, predicted data carried by feedback connections, based on a hypothesis (of perception), is compared with feedforward data (sensory input) to generate an error signal, which is carried to the next hierarchy level to generate predicted data. This process is repeated at several levels to generate appropriate predictions. A simulation of this model based on anatomical and physiological data was used to generate appropriate predictions about statistical regularities present in images, which suggested the biological plausibility of this predictive coding model [13]. Moreover, it is likely that dynamical models will gain from connections with other areas of the brain at various hierarchical levels, which would improve the prediction or increase the certainty in neural circuits about adaptive environmental constraints. Dynamic processes forming the basis of these models are consistent with the principles of ecological psychology, in that they do not require a preexisting internal model of the world, but rather depend on instantaneous representations, provided during the interactions between the organism and the environment.

However, many of the current models [9,10,11,12,13,14,15,16] do not address an important principle of ecological psychology, which is the mutual dependence between action and perception [6]. We propose that the increase in mutual information, reflecting a decrease in the total entropy in neural circuits, during the processing of sensory inputs and motor outputs, is at least partly responsible for the mutual dependence between perception and action. Entropy measures the uncertainty in the binary states (spiking versus resting) of neurons in neural circuits. An increase in mutual information reflects the decrease in the uncertainty (entropy) in the binary states of neurons in neural circuits, given the knowledge about the characteristics of environmental sensory stimuli and types of motor responses.

A reduction in entropy in neural circuits results from the interaction of the brain with the external environment, involving sensory inputs, which introduces changes in the probability laws governing the binary states of the neurons. This view is supported by the experimental data in behaving animals, which showed statistically reliable spiking patterns among specific neurons [17,18]. In the same experiment, spike patterns showed stable correlations with behavior as well as significant inter-neuronal correlations [19]. This suggests that there is an increase in mutual information, equivalently stated, an increase in certainty in the probability distributions of the binary state of neurons, given the knowledge about the interaction of the animal with the external physical environment (task). It is also noteworthy that the blood brain barrier isolates the neural circuits of the brain from unpredictable influences of neurotransmitters in blood. As a relatively isolated system, the brain becomes an open system only during interaction between the organism and the environment. Thus, the feedback and feedforward interactions, such as in predictive coding, occur independently within neural circuits. This is an important consideration as probability distributions in the binary states of neurons will be affected only by sensory inputs in addition to internal sources, but not by any other external influences. 

We also consider sparse coding, a low probability event, as a source of a large amount of Shannon information [20] available for processing perception and motor interaction. Bahmer and Gupta [21] have argued that dense coding from gamma oscillations, such as those present in lower cortical areas, can be transformed into sparse codes (low probability of activity) with the help of an integration process that involves coincidence detection by at least two other neural events, namely, low frequency oscillations that synchronize different circuits [22] and commonly observed ramping activities of cortical neurons [23,24,25]. This integration process, which generates a sparse code from a dense code via coincidence detection, would be responsible for generating the information that underlies perception. A study has shown that the conversion of dense to sparse code, representing the identity of an odorant in the nervous system of the locust, can result from coincidence detection [26]. Note that the synchronization of gamma oscillations’ communication through coherence (CTC) is also sufficient for information coding [27]. 

Information-theoretic methods are used to argue the importance of the role of mutual information and sparse coding in perception and motor interaction of the brain with physical surroundings. For this analysis, two states of neurons are considered: (a) Active state; with spikes; for simplicity there will be no consideration of the rate of spike generation or the ability of spikes to produce effects; (b) inactive state (no spikes). Circuits will be considered as probability distributions of neurons with two (binary) states: With spikes and without spikes.

We will argue that an increase in mutual information together with Shannon information play an important role in the coupling of perception and action and are eventually responsible for perception.

## 2. Methods 

### 2.1. Mutual Information between Two Successive Neural Circuits

Mutual information is a general measure of the strength of the association between two variables. Mutual information between two distributions of variables, represented by the binary states of neurons, in pairs of circuits in a network gives a measure of strength of connection in the brain. In fact, the mutual information kernel has been shown to be a good measure of functional connectivity in non-stationary data, such as electroencephalograph (EEG) [28]. 

Given two variables, *X* and *Y*, the mutual information, I(X, Y), is the average reduction in uncertainty about *X* that results from knowing the value of *Y*, and vice versa [29]. For the analysis of mutual information in neural circuits, we consider the probability distribution of random variables, defined as binary states of neurons, active (showing spike generation) and inactive (without any spikes), in two separate pools of neurons, *X* and *Y*. m and n are the number of neurons in the *X* and *Y* neuronal pools, representing two local circuits. 

The mutual information between the variables, *X* and *Y*, is defined by the following Equation (1) [29]: (1)$$I\left(X,Y\right)=\overset{m_{x}}{\underset{i=1}{\sum }}\overset{n_{y}}{\underset{j=1}{\sum }}p\left(x_{i},y_{j}\right)log\left(\frac{p\left(x_{i},y_{j}\right)}{p\left(x_{i}\right)p\left(y_{j}\right)}\right)$$

If the variables in the two distributions, *X* and *Y*, are independent, then the joint probability, $p\left(x_{i},y_{j}\right)$, will equal the product of the probability of the activity of individual neurons, that is, $\;p\left(x_{i},y_{j}\right)=\;p\left(x_{i}\right)p\left(y_{j}\right)$. In this case, mutual information, I(X,Y), will be zero since $log\left(\frac{p\left(x_{i},y_{j}\right)}{p\left(x_{i}\right)p\left(y_{j}\right)}\right)$ is reduced to log1 or zero. 

However, if a change in the binary activity state (active vs. inactive) of neurons in circuit *X* affects the probability of the binary states of neurons in circuit *Y* via direct or indirect synaptic connections or due to an external physical constraint, it will increase the joint probability, $p\left(x_{i},y_{j}\right)$. Moreover, note from Equation (1) that an increase, but not a decrease, in the joint probability, ($p\left(x_{i},y_{j}\right))$, increases the mutual information (Table 1). Furthermore, an increase in the joint probability, ($p\left(x_{i},y_{j}\right))$, intuitively implies that there is a greater certainty about the binary activity state of the variable, *y*, given the knowledge about the variable, *x*. Although, mutual information is a symmetrical quantity, it does not require reciprocal connections. Mutual information can also be computed even if there are unidirectional connections between two circuits.

The expected (*E*) or average value is defined in Equation (2):(2)$$E\left(X\right)=\;\sum_{i=1}^{m}p\left(x_{i}\right)x_{i}$$

Replacing different terms in Equation (1) with the expected (*E*) or average values gives the next equation:(3)I(X,Y)=E[log(1p(x))]+ E[log(1p(y))]− E[log(1p(x,y))]

Substituting the terms in Equation (3) with entropy values gives the following form of the equation:(4)I(X,Y)=H(X)+H(Y)−H(X,Y)

A rearrangement of Equation (4) gives another useful form of the equation (Equation (5)). This equation tells that the total entropy (H(X)+H(Y)) is reduced whenever there is an increase in mutual information, computing the net value of the joint entropy (H(X,Y)):(5)H(X,Y)=H(X)+H(Y)−I(X,Y)

### 2.2. Relative Contributions of Conditional Entropy and Mutual Information in the Total Entropy

Various neural phenomena, such as coincidental activation, synchronization by single neural oscillations, or nested oscillations, increase the joint probabilities of activation and/or inactivation of neurons, which increases the mutual information between two connected circuits of the brain. The gain of mutual information leads to the increase in the certainty in the state of neural circuits given the knowledge about the sensory stimulus. The uncertainty (entropy) in the distribution of neurons with a binary state, given the knowledge of the external stimulus, which is also referred to as noise, present alongside with inputs from the sensory stimulus, plays an important role determining how accurately mutual information is reflected in the input.

The conditional entropy, H(Y|X), is defined as the uncertainty in the data from *Y* after observing the data from *X* [29]. The conditional entropy (H(Y|X), in the data from circuit *Y* after observing the data in *X*, is defined as noise in this manuscript since this part of the total entropy in circuit *Y*, H(Y), does not give any information about the data in *X*. The mutual information, I(X,Y), which results from changes in the probability distribution of binary states in connected circuits, given the knowledge of the input source or stimulus, is an index of the signal. Mutual information, I(X,Y), is related to the conditional entropy, H(Y|X), according to Equation (6):(6)I(X,Y)=H(Y)−H(Y|X)

If this conditional entropy remains large after interacting, then there would be a larger neuronal pool that will be available for further reduction of the uncertainty, given the knowledge about the external physical stimuli (see Equations (8) and (9); Figure 1). Thus, conditional entropy, which is referred in this paper to as noise, instead of serving no useful purpose, serves as the extra capacity for additional increases in mutual information, which may result following interactions with other circuits (Figure 1).

The effect of the availability of several independent circuits in increasing mutual information can be understood in a hypothetical, but a typical situation. In this example, circuit (A), processing a given input (from primary sensory areas or higher areas), has connections with other circuits (B, C, and D) in the brain (Figure 1); the mutual information, given the knowledge of the source of inputs, can be analyzed by the following equation, by a simple addition of the mutual information between different pairs of circuits formed with circuit A:(7)$$\begin{array}{l} I\left(A,B\right)+I\left(A,C\right)+I\left(A,D\right)\\ \;=3H\left(A\right)+H\left(B\right)+H\left(C\right)+H\left(D\right)-\left(H\left(A,B\right)+H\left(A,C\right)+H\left(A,D\right)\right) \end{array}$$

The net increase in mutual information, representing the strength of the association of circuit A with circuits B, C, and D, given the knowledge about the source of the input, is quantified as I(A,B)+I(A,C)+I(A,D).

The total entropy of circuit A can be analyzed in terms of the mutual information and conditional entropy in Equation (8). The conditional entropy is the uncertainty in the data in circuit A after observing data in the other circuits, B, C, and D (Figure 1):(8)H(A)=I(A,B)+I(A,C)+I(A,D)+(H(A|B)+H(A|C)+H(A|D))

If another circuit, E (dotted circle), processing the same inputs, begins to interact with circuit A, then it would increase the mutual information, given the knowledge about the characteristics of the input. In other words, this additional interaction with circuit E will further increase the certainty in circuit A given the knowledge about the source responsible for inputs. This would relate physiologically to increasing the attention to the source of inputs. The increase in mutual information will come from the total entropy of circuit A (H(A)), which is evident in Equation (9):(9)H(A)=I(A,B)+I(A,C)+I(A,D)+I(A,E)+ (H(A|B)+H(A|C)+H(A|D)+H(A|E))

Since the conditional entropy, (H(A|B)+H(A|C)+H(A|D)+H(A|E)), is the uncertainty in the data in A after data in B, C, D and E are observed, it represents the portion of the total entropy of A that does not provide information about inputs processed by the network, now referred to as ‘noise.’ On the other hand, ‘signal’ is the reduction in uncertainty, a correlate of perception, in circuit A after observing data in B, C, D and E, which is the increase in mutual information, I(A,B)+I(A,C)+I(A,D)+I(A,E). Thus, the signal to noise ratio can be analyzed by Equation (10):(10)$$\frac{{Signal}^{\prime }}{{noise}^{\prime }}=\frac{I\left(A,B\right)+I\left(A,C\right)+I\left(A,D\right)+I\left(A,E\right)}{(H\left(A|B)+H\left(A|C\right)+H\left(A|D\right)+H\left(A|E\right)\right)}$$

Equation (10) can be rewritten in another form (Equation (12)), which expresses conditional entropy as the difference between the total entropy and mutual information. As explained later, the signal to noise ratio will increase whenever an additional circuit couples with the circuit, such as circuit E to the already networked B, C, and D. The networking of another circuit E will then increase the mutual information by I(A,E) (Figure 1), which is done by subtracting from the total entropy of circuit A that constitutes the conditional entropy, after data in B, C, and D are observed. Notice that if there is an increase in the joint probability of pairs of neurons due to an increased volume of the input served by the same sensory pathway, then it would increase the mutual information in neural circuits, related to the source of the sensory input. The increase in mutual information would increase the signal to noise ratio according to Equation (11):(11)$$\frac{Signal}{noise}=\frac{I\left(A,B\right)+I\left(A,C\right)+I\left(A,D\right)}{H\left(A\right)-\left(I\left(A,B\right)+I\left(A,C\right)+I\left(A,D\right)\right)}$$

Once circuit E, also processing the same stimulus or internal source, is added to the network, the signal to noise ratio is given by the following equation:(12)$$\frac{{Signal}^{\prime }}{{noise}^{\prime }}=\frac{I\left(A,B\right)+I\left(A,C\right)+I\left(A,D\right)+I\left(A,E\right)}{H\left(A\right)-\left(I\left(A,B\right)+I\left(A,C\right)+I\left(A,D\right)+I\left(A,E\right)\right)}$$

Comparison of Equations (11) (circuit E is not in the network) and (12) (circuit E is in the network) (inequality Equation (13)) reveals that the addition of another circuit (circuit E), controlling the same task guided by the same stimulus, to a network will increase the signal to noise ratio. Since the signal (mutual information) is a direct correlate of perception, the increase of the signal to noise ratio will result in increased attention to the perceptual object (Table 1). This result is used for arguing that perception, which is partly dependent on the signal to noise ratio, is the result of the networking of many circuits to a common circuit (circuit A) that primarily processes an input:(13)$$\frac{{Signal}^{\prime }}{{noise}^{\prime }}>\frac{Signal}{noise}$$

Therefore, it will be advantageous for the entropy of circuit A, H(A), to be a large quantity, so that there is more capacity for an increase in mutual information, which would allow more complex processing of perceptual objects. Thus, it is likely that the best candidates for circuit A would be network hubs, and association areas, which include large inter-connected parts of the cortex [30,31].

Note that an important condition to be present for Equations (7)–(12) to be true is that the circuits of A, B, C, D, and E are part of the same network, interacting with the same internal source or external stimulus. Although not independent, since they process the same input source, B, C, D, and E are assumed to be probability distributions that are formed by separate sets of variables, namely the neurons with binary states. 

### 2.3. Surprise, Entropy, and Shannon Information

Unlike a coin toss, classification of neural spikes as a binary event is less obvious. Spiking or action potential represents one binary state; the other state would be represented by the absence of the action potential, or when the neuron is in the resting state. Like the toss of coin, a spiking neuron shows two outcomes during its activity, namely the presence or the absence of spikes at a particular time. Moreover, the generation of a neuronal spike is determined by a probability function. Shannon information from observing one event/outcome is the measurement of the average surprise associated with that particular event/outcome [20,32]. In case of neurons, the one particular outcome of physiological interest will be the generation of spikes. In contrast, entropy is the average surprise after observing all outcomes [20,32], which are both, spikes and no spikes (resting state). 

Shannon information or average surprisal [20] after observing spike generation in a population of m neurons in an area of the brain is given by Equation (14); p(X) is the probability function that governs the spiking activity of the neuron:(14)$$Shannon\;Information\;=\;-\sum_{i=1}^{m}p_{i}log\left(p\left(X\right)\right)$$

Sparse coding refers to information coded by infrequent activities in a small number of neurons from a larger set [33]. Sparse coding is associated with apparently complex motor outcomes in song birds, such as highly stereotyped songs [34,35]. Neurons that are responsible for sparse coding exhibit a low average probability of firing [33,34,36]. Thus, according to Equation (14), Shannon information after observing a spike generation in a sparsely firing neuron will be a large quantity. Despite large amounts of Shannon information that result from a low probability event, it is not clear how sparse coding is mechanistically related to complex stereotyped outcomes [33]. 

As depicted in Figure 2, sparse coding could result from coincidence detection by the active phase of a low frequency oscillation and the stage of climbing neuronal activity. If the excitatory phase is a fraction (1/k) of the total length of the oscillation cycle, the probability of finding a low frequency oscillation in the excitatory phase that undergoes a sensory stimuli-induced random phase-shift [37] will be 1/k. Additionally, the probability (p(L)) of ramping neurons reaching a threshold will depend on the probability function governing slope (L) of the ramping activity. Also note that for the purpose of analysis of the ramping activity in neural circuits, the appearance of the excitatory phase of a stimulus-induced phase-shifted low-frequency oscillation leading to a coincidence detection is a random event. When coincidence detection by two relatively independent events occurs, it will lead to increased activity in a set of neurons, receiving inputs from ramping neurons, during the excitatory phase of reset low-frequency oscillation (Figure 2). The probability (p(S)) of a sparse outcome, due to the activity of ramping neurons, is given by the following equation:(15)$$p\left(S\right)=\frac{p\left(L\right)}{k}$$

Note that the term, $\frac{1}{k}$, expressing the probability of finding the oscillation in the excitatory phase, in Equation (15), is an approximation, with the assumption that the excitatory phase is a single discrete event. However, the probability of spike generation during the excitatory phase of oscillation would vary according to a continuous trigonometric function. 

Furthermore, according to the model suggested above (Figure 2), the information that would be produced by sparse coding is a consequence of two relatively independent events, ramping activity, representing internal states [38] and a stimulus-induced phase shift in neural oscillations. Thus, Shannon information generated by this mechanism enables the brain to provide optimal outcomes during an interaction with the external surroundings. Furthermore, the independence of two events will provide a limited number of ramping activities the ability to interact with various neural oscillations that are randomly phase-shifted by sensory stimuli [37] to generate large amounts of Shannon information in a very wide range of brain functions. 

Ramping activities, which are the most common patterns in the frontal cortex during timing tasks, are shown to be important in the temporal control of actions [23,24,25]. Furthermore, a study in rats suggests that ramping activities in the orbitofrontal cortex represent internally generated waiting control [38]. 

The proposed role of ramping neuron activities in information processing is consistent with its role in distributed modular clocks, proposed by Gupta [39]. Sparse activity can result from the interaction with the environment based on the postulated mechanism since a variety of stimuli, such as visual, auditory, and proprioception, can cause random phase-shifting of neural oscillations. Note that the random nature of stimuli onset during interaction with the environment is due to the lack of knowledge about external stimuli in neural circuits before their onset. The observation of sparse activity can be quantified as surprisal or Shannon information (Equation (14)), which would potentially encode behavioral outcomes, such as timing. 

A study of Shannon information in imaging systems has shown that Shannon information to some extent rises with an increase in the signal, measured as the image diameter or threshold of detection after parameter optimization [40]. This suggests that large amounts of Shannon information could also encode complex functions of the brain for processing working memory functions or for generating episodic memories. Moreover, note that dense coding, in contrast to sparse coding, will interfere with the increase in mutual information. 

## 3. Summary of Important Results

Important Results are summarized in Table 1.

**Table 1 entropy-21-00365-t001:** Important Results.

1. Mutual information, I(X,Y) = 0, if activities of individual neurons are independent events (Equation (1))
2. An increase, but not a decrease, in the joint probability, $p\left(x_{i},y_{j}\right),$ increases the mutual information (Equation (1))
3. Increase in mutual information, from an increase in the traffic of sensory inputs, served by the same pathway, will increase the signal to noise ratio (Equation (11))
4. Addition of another circuit guided by an action to a network will increase the signal to noise ratio (inequality Equation (13))
5. Areas with sparse activity associated with a low probability of neuronal activities generate large amounts of Shannon Information (Equation (14))

## 4. Discussion

### 4.1. Effect of Temporal Coupling of Neural Activities on the Measurement of Mutual Information

We have argued, based on Equation 1, that an increase, but not a decrease, in the joint probability ($p\left(x_{i},y_{j}\right))$ increases the mutual information (Table 1). If there is simultaneous activity of neuron pairs (*x_i_* and *y_j_*) from separate neuronal pools (*X* and *Y*), either as a result of a task-dependent constraint or synaptic connections, then this would lead to an increase in joint probability. In one scenario, there is a simultaneous activation of pairs of neurons in two neuronal pools, via chemical or electrical (gap junction) synaptic connections, which will increase the probability ($p\left(x_{i},y_{j}\right))$ of the joint activity of neurons, *x_i_* and *y_j_*. In the other scenario, pairs of neurons are activated simultaneously because they are both controlled by a specific aspect of the external stimulus. 

### 4.2. Web-Like Configuration following Processing of Specific Inputs Imposes Patterns of Activation

Activation of neurons in networked circuits, as inputs are being processed, would lead to a web-like configuration of circuits (Figure 3), where dots represent neurons, and the (double arrow) lines connecting the dots represent the states associated with an increased probability of joint activity of pairs of neurons. An increased joint probability of activity will reduce the total entropy by an amount called mutual information, given a specific stimulus (*), if the network processes the same stimulus (*) (Equation (1), result 2). The increase in the amount of mutual information, which is quantitatively related to the increase in the joint probability of activity of pairs of neurons in two circuits, is correlated to action and perception. Connected dots, representing pairs of neurons with a high probability of joint activity, will also contribute to the reduction in total entropy in other circuits, where they will increase the certainty about the environmental sensory model, further contributing to the certainty related to a perceptual object or a specific motor response. 

### 4.3. Increase in Mutual Information Contributes to Perception

Here, we argue that the presence of robust increases of mutual information is the crucial link for perception, which is mainly based on experimental and theoretical evidence available from an increasing body of research data in the processing of visual information. It was proposed earlier by von der Malsburg [41] that neuronal responses can be synchronized for processing the grouping of stimulus-specific features of perceptual objects. The experimental evidence supporting the role of synchronization in the temporal coupling of neuronal responses in processing stimulus-specific features were provided later by the Singer laboratory [42,43,44]. Neuenschwander and Singer [43] demonstrated that temporal coupling among responses of spatially segregated ganglion cells can be exploited to convey information relevant for perceptual grouping [20]. Bahmer and Gupta [20] have recently argued based on the studies of other perceptual functions, including auditory, olfactory, and interval-timing, that temporal correlations between neural events form an important basis of perception. 

### 4.4. Retinal Processing of Visual Information

We propose that the mutual information, a measure of the association between two discrete spatial points, will determine the perception of how close two spatial points from each other are. Furthermore, since there is direct electrical coupling formed by the gap junctions between the rods and cones in different animals, including primates [45,46,47], there will be an increase in the joint probability of their activation, increasing mutual information.

An increase in the joint probability of activation of the rods and cones, via direct or indirect coupling, will also increase the joint probability of their activity of their cortical connections, serving distinct spatial points on the physical object. In addition to the gain in mutual information (Table 1), this would lead to a dimensionality reduction by converting multiple spatial points of physical object(s) into a single neural representation for the purpose of feedback loops. This will reduce the information processing load, especially for those functions that are dependent on feedback processes, by eliminating the need for multiple independent feedback loops. Moreover, the representation of multiple spatial points of a physical object, as a single neural entity in motor feedback interactions or during working memory functions, will also increase mutual information, which would improve the signal to noise ratio, contributing to the attentional modulation as well as the perception of the physical object (see Equations (11)–(13)). 

### 4.5. Perception of Physical Continuity

Ganglion cells in the retina are coupled together by direct as well as indirect gap junctions [48]. Direct and indirect coupling of ganglion cells increases the joint probability of activity, which will depend on the number of gap junctions connecting the pair. The effect of an increased joint probability of activity of ganglion cells will also be extended to the corresponding pairs of third order neurons, which project to the primary visual cortex. When specific pairs of third order neurons show an increased joint probability of activity, spatial points served by the corresponding photoreceptors will be perceived as being physically closer; or continuous if the joint probability of activity approaches 1.

Furthermore, the mutual information processed from the rods and cones will also be affected by other complex interactions within the retinal circuitry as reviewed recently [47], which could serve as a basis of color, depth, or texture perception.

### 4.6. Success of Motor Interaction Depends on a Decrease in Entropy Given the External Sensory Objects

A successful motor interaction of the brain with external objects requires a robust increase in mutual information given the knowledge of sensory stimuli. This will result from an increased number of pairs of neurons with a higher probability of joint activity, leading to a web-like connectivity pattern (Figure 3). This increase in certainty will set the stage for a successful motor interaction with the environment [39,49,50]. 

In a past study, gamma oscillations, localized in the primary motor cortex, were seen to reach a peak amplitude during the movement [51]. The same study also showed that gamma oscillations were absent during the sustained part of isometric movements, when there is no finger movement or muscle shortening [51]. It is interesting to note that the presence of movements, which involves an interaction with the environment, shows a gamma band peak in the primary motor cortex, while there is no increase in the amplitude of gamma bands when there are no movements, consistent with the lack of direct interaction with external surroundings. Robust increases in mutual information between successive circuits, from sensory areas to the premotor areas, result in movements that produce successful interaction with the environment. In fact, several studies have shown the presence of coherence, consistent with a decrease in uncertainty via gamma band synchronization, between different areas involved in visuo-motor transformations, starting from the early visual areas and reaching through the parietal cortex and motor cortex to the spinal cord [27]. Thus, gamma oscillations by increasing mutual information may play a key role in cognitive functions. Hence, it is not surprising that different gamma oscillation activities are also found to be reduced in schizophrenia, which is primarily a disorder of cognitive functions [52]. 

### 4.7. Robust Reduction of Entropy, Given Stimulus Characteristics, Involves Many Parts of the Brain 

The two visual pathways hypothesis suggests the processing of visual information by two distinct systems in the brain: (a) The dorsal stream for the visual control of skilled actions and (b) the ventral stream for the identification of objects [31,53,54] (Figure 4). The dorsal stream runs from the primary visual area to the middle temporal area (MT) to posterior parietal areas, projecting to the premotor areas [31,54,55]. The ventral stream projects from the primary visual area to the inferotemporal area [31]. There are extensive connections between both pathways [31], which suggests that there will be robust increases in mutual information dependent on both the visual and motion-dependent characteristics of visual objects. Note that the MT is specialized to process velocity, direction, and depth [55]. Furthermore, the cerebellum, which depends on feedback mechanisms for the calibration of time-representation in neural circuits [39], is connected to the cerebrum by multiple parallel loops; that is, the cortical areas project to the same part of the cerebellum, from which they receive inputs [56]. Interestingly, although the cerebellum has major connections with the parietal and prefrontal cortices, it also appears to have reciprocal connections with the temporal lobe in humans [56,57,58,59]. Due to the connections of the cerebellum with many parts of the dorsal stream, it is likely to be responsible for large increases in certainty in the brain circuits about objects in visuomotor tasks during the motor control of visual objects. This increase in mutual information will increase the signal to noise ratio according to Equations (11)–(13), contributing to attention and other aspects of action-perception related to visual objects. Moreover, a past study suggests that a single representation guides both action and perception, which is consistent with extensive interconnections between dorsal and ventral streams [60].

A study of a visuomotor task, using a trackball to manipulate a randomly rotating cube on a computer screen, revealed the presence of coupling between the phase of delta oscillations (2–5 Hz) and an increase in the amplitude of gamma oscillations (60–90 Hz) in the occipital and parietal cortical areas as well as the cerebellum [61]. Based on the role of gamma oscillations in producing tighter temporal coupling of synaptic inputs [27], referred to as communication through coherence, this increase in the amplitude of gamma oscillations is consistent with increasing mutual information through the synchronization of inputs into various hubs of the network that form the dorsal stream (Figure 4).

Figure 4 depicts the circuits of the dorsal and ventral streams, illustrating that the interactions between successive circuits lead to the increase in mutual information. One can compare the inferotemporal area in Figure 4, which is a major hub interacting with other brain areas during visuomotor tasks, to circuit A depicted in Figure 1. Note that the inferotemporal area, which forms the ventral stream, has connections with the primary visual area, posterior parietal area, and cerebellum. Large numbers of interconnections of the inferotemporal area effectively increase the total entropy, which makes it suitable for large increases in mutual information required for attention to visual objects during visuomotor tasks [61]. 

### 4.8. Reduction in Joint Entropies via Interaction Between Feedforward and Feedback Connections in the Cortex

In order to interpret sensory data based on the Bayesian scheme, the brain may use several constraints, such as, resulting from prior experience, recent experience, present data, and an internal model of the world. Predictive coding specifically refers to the use of an internal model to interpret sensory data [9,13,62]. Feedforward connections carry current expectations of sensory data while feedback connections carry optimal expectations of sensory data [13,62]. Feedforward connections, which are synchronized by high-frequency gamma oscillations, originate predominately from the superficial layers while feedback connections, carried in the deep layers, are synchronized by alpha and beta oscillations [63,64], consistent with processing in networks [22] that would allow a decrease in the total entropy (Figure 4). An increased number of pairs of neurons with higher joint activity, resulting from feedforward–feedback interaction, which leads to a reduction in entropy, will form an instantaneous representation of organism–environment dynamics. 

### 4.9. Increase in Joint Probability of a Group of Neurons: A Result of External Physical Constraint

If a visual stimulus is present in the receptive fields of a specific set of neurons, then all neurons in this set would become active simultaneously, even if those neurons are not connected by synapses or gap junctions. In this case, the temporal correlation of neuronal activities is not due to the presence of synapses, but it is due to an external constraint, namely, the simultaneous presence of stimuli in the receptive fields of these neurons. Such temporal coupling was observed in a study in which neurons in early visual areas fired synchronously at 40 Hz (V1 and V2) when the visual stimulus was simultaneously present in their receptive fields [42]. Similarly, during a motor interaction, motor circuits for common muscles may be activated together in both hemispheres without mutual synaptic connections. Again, this is the result of an external constraint. 

### 4.10. Role of Phase-Resetting of Neural Oscillations in the Temporal Processing of Information 

Sparse coding, which codes sensory and motor information, results from the synchronous activity of neurons in a large area of the brain [33]. In one of the plausible mechanisms, the probability function controlling sparse activity will depend on the low-frequency oscillations, the phase of which can be reset by a sensory stimulus or internal cue [37,65]. According to the modular clock mechanism proposed by Gupta [39], timing information in neural circuits is calibrated as a result of sensory and motor interaction with the physical surroundings. Stimulus-induced phase resetting is likely to be an important mechanism responsible for the calibration of timing information in neural circuits during sensory interaction with the physical environment. Moreover, stimulus-induced phase-shifting of low-frequency oscillations will affect the timing as well as the amount and pattern of Shannon information generated, playing an important role in producing a favorable outcome during an interaction with the environment. 

A past study had shown synchronization of neuronal activity in the visual areas of the cat when presented with optimally aligned bars in corresponding receptive fields [42], which is consistent with a stimulus-induced phase shift. The stimulus induced phase-synchronization, a neural event corresponding to the presentation of a stimulus, would lead to the generation of information according to the mechanism that involves ramping activity as outlined above (Figure 2). This would associate the time of an external event, presentation of stimuli, with the time of generation of Shannon information, creating a representation of the physical time-coordinate. The creation of an internal representation of the physical time-coordinate, plotted by the processing of several sensory stimuli at separate physical time-points, will help to input physical time information into neural circuits that process timing functions as proposed by Gupta [39]. Arguably, the generation of Shannon information in separate circuits, processing different features synchronously, is responsible for the feature binding resulting from stimulus-induced synchronization independently in small areas of the brain even when the EEG is desynchronized [66].

### 4.11. Perceptual Cycles and Information Processing

A large body of evidence now suggests that perception in various modalities is discrete rather than continuous in nature [67,68,69]. Experimental data have shown that low-frequency oscillations adjust the excitability of neurons, which is measured as high-frequency oscillations [70], across the occipital, parietal, and frontal regions, which can predict behavior on a sub-second time scale [71]. The spectral analysis of modulations in perception and attention in different modalities show that attention and perception are modulated with ~7 and ~10 Hz frequencies, respectively [68]. The detection of the perceptual cycle supports the suggestion that there is a temporal correlation of neurons during the excitatory phase of neural oscillations. 

Phasic increases in mutual information, which represent certainty about input signals generated internally or from an external stimulus, are consistent with a recent study that shows that the visuospatial attention is associated with robust and sustained long-range synchronization of cortical oscillations exclusively in the high-alpha (10–14 Hz) frequency band, connecting frontal, parietal, and visual regions, and was observed concurrently with the suppression of low-alpha (6–9 Hz) band oscillations in the visual cortex [72]. It is noteworthy that higher frequency oscillations in the beta range (or high alpha range) promote local gamma synchronized activities, increasing mutual information, and low alpha range oscillations inhibit local gamma synchronized activities [27]. Therefore, this study [72] supports the role of the frontoparietal network in generating mutual information in discrete time intervals during the active phase of high range alpha oscillations (10–14 Hz range), which is responsible for visuospatial attention. Thus, phasic increases in mutual information, mechanistically supported by high-frequency oscillations nested in low-frequency oscillations, are likely to be responsible to the modulations in behavior in visuomotor tasks. Anatomical data also suggest the presence of structures throughout the brain that can serve as the basis of producing nested oscillations [73]. There are several parts of the brain that contain helix-like anatomical structures, resolved at the level of cells, consistent with hierarchical processing. The examples include the ventral part of the lemniscus lateralis, locus coeruleus, oculomotor nuclei amygdala, hippocampus (cornu ammonis 3), and pars compacta and reticulata of the substantia nigra [73].

### 4.12. Affordance and Mutual Information

The above analyses of mutual information explain how affordances relate to the organism, information, and event. Affordances are perceivable environmental opportunities for behavior or action possibilities [1]. Affordances depend upon the properties of the organism and the environment, for example, if an organism is tall (property of the organism) enough to jump a fence (environment). As depicted in Figure 4, different areas of the brain in the dorsal and ventral streams, which guide perception and action, respectively, interact as they are interconnected. This interaction, between dorsal and ventral streams, will increase the probabilities of joint activity of select neurons in different parts of the brain, leading to an increase in mutual information (Table 1), given (i) visual stimuli from environment (fence) and (ii) motor response (jump) to visual stimuli. A recent study by Josa et al. [74] showed that the perception of the distance of a target in front of subjects who pushed a trolley varied, which depended on whether the trolley was heavy (loaded with book) or light (empty). Their results show that the perceived distances were greater when the trolley was loaded. These results agree with Gibson’s ideas about perception. Gibson argued that the primary purpose of perception is to guide action and not to gather information about the environment [1,5]. In this example, the addition of the trolley adds another visual stimulus source, which would lead to joint activity of a new set of neurons, which would increase the mutual information related to visual information (object properties in affordance). The increase in mutual information will increase the signal to noise ratio (result 3), which would enhance attention to action opportunities; that is, to move the trolley to the target. 

### 4.13. Action-Perception Involves a Combined Increase of Mutual Information and Shannon Information

As defined before, given two variables, *X* and *Y*, the mutual information, I(X, Y), is the average reduction in uncertainty about *X* that results from knowing the value of *Y*, and vice versa. Since mutual information is the increase in the certainty about a stimulus, given the knowledge about the stimulus, this measurement corresponds to a representation of opportunities for action. In contrast, a Shannon information increase is related to the evidence for the sensory model. This can be argued with the help of a hypothetical example of a photon source from a visual object. In this example, the photon source from a visual object repeatedly stimulates a specific photoreceptor, which is an increase in Shannon information as the photoreceptor cell would not fire or occasionally fire if that photon source was absent. If there is a stronger stimulation of the photoreceptor, then there will be a greater increase in the surprise and more robust evidence of the visual object. The increase in Shannon information is directly related to the evidence of the photon source or visual object. Furthermore, the formula (Equation (14)) for Shannon information assumes that there is no previous knowledge (or internal representation) of the visual object in neural circuits. Now, we argue that perception and action is the result of combined increases in mutual information, representing opportunities for action, and the increase of Shannon information, which represents evidence of the sensory model of the environment (Figure 5). The combined changes in mutual information and Shannon information help perception to guide the action. Perception emerges from the coupling between action and perception (Figure 5). 

### 4.14. Self-Organization of Cognitive Structures in Action-Perception

For the smooth guidance of action by perception, it is important that the constraints forming old internal cognitive structures are replaced with the constraints, paving the way for self-organization of new cognitive structures. Self-organization, leading to the formation of cognitive structures, can be analyzed by the study of thermodynamic entropy [75]. Entropy in thermodynamics is a measure of disorder in physical states and is related to information theoretic entropy by a minimum energy cost of erasing or acquiring one bit of information [76,77]. 

The evidence for the resolution of internal cognitive structures, by replacing old constraints with new constraints, is provided in a past study employing a thermodynamic model, in which subjects solved gear-system problems [78]. In the gear-system problem paradigm, subjects were asked to predict the movement of a target mechanical gear, given the turning direction of a driving gear, which were projected on a computer screen. Stephen, Dixon, and Isenhower [78] measured the time series of finger motion, and then by using recurrence quantification analysis, they studied the self-organization process, responsible for the formation of internal cognitive structures. Recurrence quantification analysis assumes that if two variables in a complex system (finger motion and self-organization of internal cognitive structures) are interrelated, then the measurement of recurrences in one variable (finger motion) will reveal the time series of the other variable (internal cognitive structures). Classical thermodynamics predict that the addition of entropy would lead to the formation of new cognitive structures. In accordance to the prediction, the introduction of perturbations in the task was shown to initially increase entropy, which was followed by a decrease in entropy, consistent with the self-organization of a new cognitive structure in a complex system, which then predicted early discovery by subjects [78]. In complex systems, the increase of entropy dissolves old constraints. Although old constraints in this experimental paradigm are likely related to the representation of action opportunities before discovery, it will require further studies to determine if the constraints responsible for self-organization of internal cognitive structures are caused by an increased probability of joint activity of pairs of neurons (Figure 3). Moreover, by increasing the task variability, that is entropy, old thermodynamic constraints are dissolved to form new thermodynamic constraints (updating the internal knowledge about the mechanical gears). This allows new cognitive structures to form, which is responsible for early discovery. Moreover, this explains why introducing variations in the task leads to early discovery in gear-solving problems [78]. 

A key challenge for understanding the dynamics of cognition has been capturing fine-grained, moment-to-moment changes in mental activity. Power-law behavior is another key indicator that can help in understanding finely-grained changes in mental activities [75]. Cognitive structures can be considered as being made of progressively smaller elements, which show same dynamics at each level of organization, called power-law behavior. When constraints holding the cognitive structures break down to allow new cognitive structures to form, then the power-law dynamics dominate according to new interactions, which leads to the formation of new cognitive structures. After the cognitive structures have reconfigured, power-law behavior decreases, and constraints hold the new cognitive structure according to updated interactions with the environment. 

## 5. Summary 

In the beginning, we introduced three main principles of environmental psychology, namely: (i) Perception is direct; (ii) perception is for action; and (iii) perception is of affordances [5]. In this article, we used the information theoretic approach to provide a deeper understanding of the last two principles. We argued that there are robust increases in mutual information between successive circuits across the brain during an interaction with the environment, which is due to the increase in joint probability distributions of binary states of neurons, given the knowledge about sensory stimuli and motor tasks (Figure 4). Robust increases in mutual information reflect the increase in certainty in neural circuits about the knowledge of an environmental opportunity for action, also called affordance. In addition, there is an increase in surprise measured by Shannon information, which is the measure of evidence of sensory objects. In combination, as depicted in Figure 5, mutual information and Shannon information are responsible for guiding action. The guidance of action results in perception, as proposed by Gibson [5]. However, this principle, namely perception, is direct, and cannot be directly supported by the theoretical work presented in this paper.

In addition, the theoretical analysis presented in this paper helps draw other important conclusions in understanding the interaction between an organism and the environment. The increase in mutual information also places constraints on how neural circuits in other areas may be activated, which would lead to specific motor responses, optimizing the outcomes of external tasks, needed for survival. Furthermore, mutual information is a symmetrical quantity, which means that if stimulus-biased probability laws increase the certainty of binary states in sensory circuits, it also results in increased certainty in motor circuits at higher hierarchical levels, given the knowledge about the stimulus, which would lead to stimulus-specific motor responses. Additionally, vice versa will also be true, a change in the certainty in motor circuits given the nature of the motor task will change the probability of joint activation of neurons between motor and sensory circuits (Figure 3), which can cause an effect on perception due to the changes in the nature of the motor task. 

Another model of perception that uses the information theoretic approach is the free energy minimization model [15,16]. Free energy in this model is an information theoretic quantity, which is the upper bound on surprise. The minimization of free energy also minimizes the surprise. In this model of perception, free energy measurement has access to two components, sensory states, resulting from sensory inputs from an agent (sensory objects), and recognition density that is encoded by neuronal activity and connection strengths in the circuits of the brain. The recognition density is a probabilistic representation of the model causing a particular sensation. By minimizing free energy, the surprise is reduced and the evidence for the sensory object is increased, which leads to perception. 

## Figures and Tables

**Figure 1 entropy-21-00365-f001:**
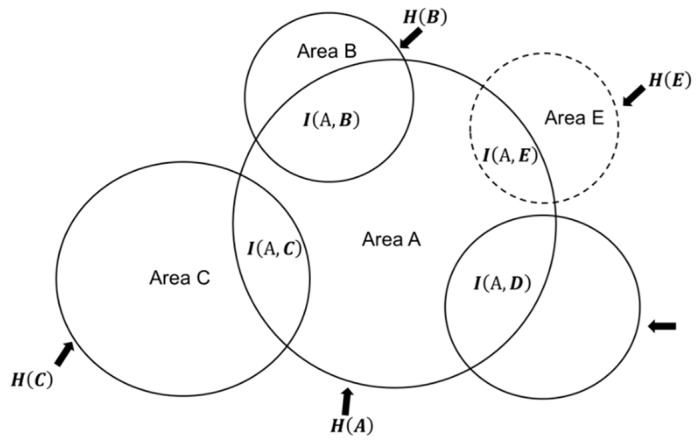
This schematic depicts a typical configuration of processing hubs formed by cortical areas in the brain. A large cortical area, A, is shown to interact with several other circuits (B, C, and D) by sending and/or receiving inputs when processing from inputs related to a common stimulus or input source. In a task with a greater complexity, another circuit, E (broken line), is shown to be involved. The addition of another circuit, to the network processing a given stimulus, will decrease the conditional entropy (area of circle A, excluding overlapping regions) and increase the mutual information (area of all overlapping regions within circle A), which will improve the signal to noise ratio (Equations (11)–(13)).

**Figure 2 entropy-21-00365-f002:**
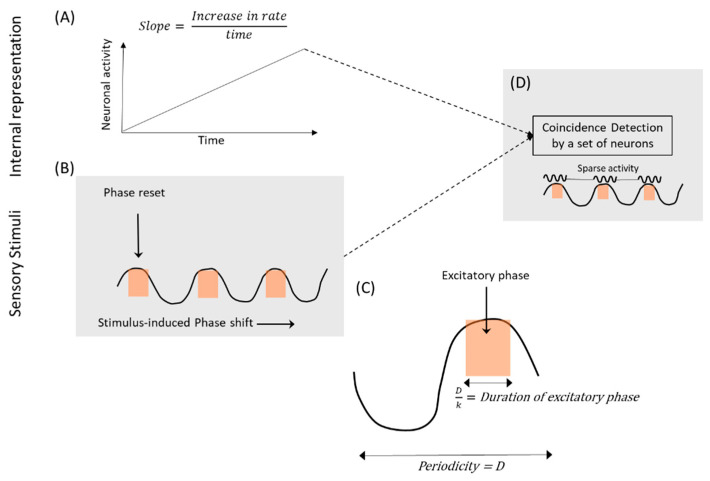
This schematic (adapted from Bahmer and Gupta [21]) depicts the coincidence detection (D) leading to the activation of a set of neurons. This sparse set of neurons are activated (D) when the excitatory phase of low frequency oscillation (B) coincides with gamma synchronized synaptic discharge [27] from ramping neurons (A). Sensory stimuli reset the phase of low-frequency oscillations, leading to a random shift in the excitatory phase (orange shaded area) of the oscillations in relation to the ramping activities, which represent internal states of the brain. Based on the fraction of the excitatory phase of the total cycle length, the excitatory phase of the low-frequency oscillation will coincide with a high-level firing state of a ramping neuron according to a probability value of 1/k. The surprisal or Shannon information from the coincidence detection, which results in the firing of a set of neurons (sparse coding), encodes the sensory and motor interaction with external environment.

**Figure 3 entropy-21-00365-f003:**
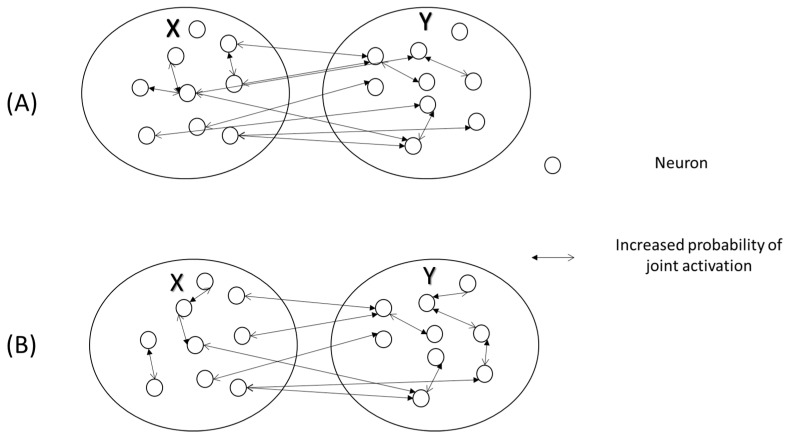
This schematic depicts web-like connectivity resulting from an increased joint probability of activity $\left(p\left(x_{i},y_{j}\right)\right)$, (shown as double-headed arrows) of pairs of neurons in the circuits, *X* and *Y*. Notice that the web-like configuration in A and B are different, which depicts different consequences due to differences in stimuli, affecting the probability laws that govern neuronal activities. Moreover, these differences in web-like patterns in two scenarios can result in differences in the outputs, which would be responsible for differences in motor or behavioral outcomes.

**Figure 4 entropy-21-00365-f004:**
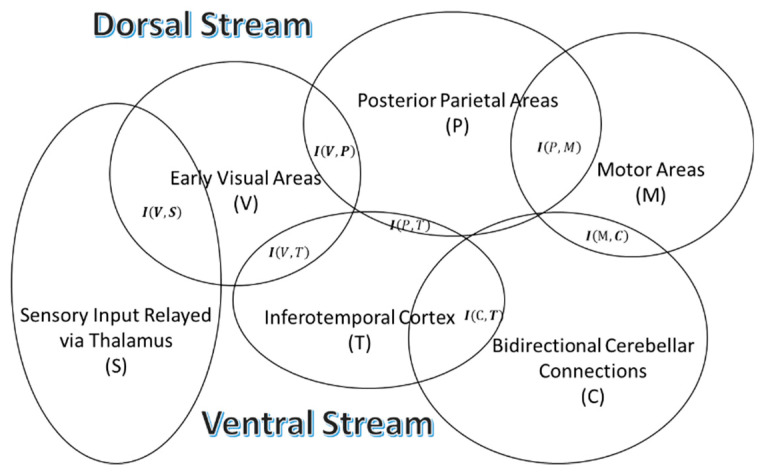
This schematic depicts interactions between different regions that construct dorsal and ventral streams, which serve as large populations of neurons with binary states, providing a large entropy. A large entropy can allow robust increases in mutual information via interactions with many circuits, helping to improve the signal to noise ratio (Equations (11)–(13)). Also notice that many regions of the dorsal and ventral streams have extensive connections with one another and with the cerebellum.

**Figure 5 entropy-21-00365-f005:**
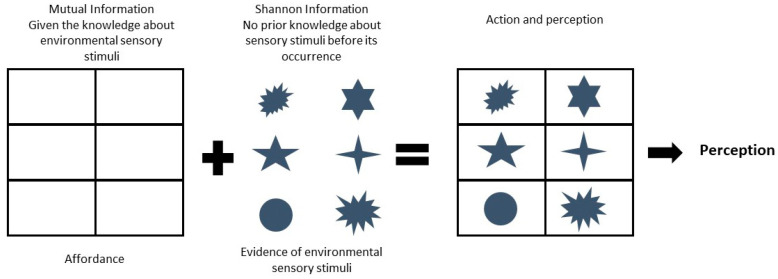
This schematic illustrates the relative roles of mutual information and Shannon information in action and perception. Mutual information is the increase in certainty in neural circuits given the knowledge about environmental sensory stimuli, and therefore it represents the environmental object properties in perception and action, also called affordance (shown as an empty frame). Since for Shannon information, one assumes that there is no previous knowledge about the sensory stimuli, its increase serves as the evidence of the source of the sensory stimuli (shown as several symbols). Both mutual information (affordance) and evidence of the sensory stimuli are responsible for action and perception. The perception results from the guidance of action by perception (embedded cognition).

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
