# Peer review of "Increase in Mutual Information During Interaction with the Environment Contributes to Perception"

_entropy, 2019, doi:10.3390/e21040365_

Round 1

Reviewer 1 Report

The manuscript is a review article about perceptual processing from a computational neuroscience perspective that reads somewhat like a collection of snippets from textbook materials without any original empirical data or results presented by the authors. I am not convinced about the novelty of the content provided in the paper. The manuscript would be much strengthened if it included discussion from competing theoretical approaches such as anti-representationalist, and/or ecological theories of perception. The usage of mutual information and entropy should be couched in some broader context of dynamical systems theory and analyses to disambiguate the measures from others that have the same labels. For example, in Recurrence Quantification Analyses, there are variables called Average Mutual Information (AMI), entropy, and percent determinism that are used to describe time series data of various motor behaviors (hand gestures, free wielding, etc.). How do those analyses and methods compare to the ones mentioned in the current manuscript? As context, here are some empirical studies using these measures:

Stephen, D. G., Dixon, J. A., & Isenhower, R. W. (2009). Dynamics of representational change: Entropy, action, and cognition. Journal of Experimental Psychology: Human Perception and Performance35(6), 1811.

Stephen, D. G., & Dixon, J. A. (2008). The self-organization of insight: Entropy and power laws in problem solving. The Journal of Problem Solving2(1), 6.

And for a review of nonrepresentational approaches to perception and action that do not assume modular systems I recommend:

Chemero, A. (2013). Radical embodied cognitive science. Review of General Psychology17(2), 145-150.

There are also doubts whether the distinction between dorsal and ventral streams is useful at all for understanding certain perceptual tasks:

Durgin, F. H., Hajnal, A., Li, Z., Tonge, N., & Stigliani, A. (2010). Palm boards are not action measures: An alternative to the two-systems theory of geographical slant perception. Acta Psychologica134(2), 182-197.

Even though I welcome and appreciate the authors stating that the dorsal and ventral systems have quite a bit of overlap, I still think it would be useful to include some of these references as contrasting views so that the reader has an appreciation that not all statements mentioned in the manuscript are settled facts, rather they are still subject to much empirical testing and theoretical debate.

Author Response

The manuscript would be much strengthened if it included discussion from competing theoretical approaches such as anti-representationalist, and/or ecological theories of perception. The usage of mutual information and entropy should be couched in some broader context of dynamical systems theory and analyses to disambiguate the measures from others that have the same labels. For example, in Recurrence Quantification Analyses, there are variables called Average Mutual Information (AMI), entropy, and percent determinism that are used to describe time series data of various motor behaviors (hand gestures, free wielding, etc.). How do those analyses and methods compare to the ones mentioned in the current manuscript? As context, here are some empirical studies using these measures:

Stephen, D. G., Dixon, J. A., & Isenhower, R. W. (2009). Dynamics of representational change: Entropy, action, and cognition. Journal of Experimental Psychology: Human Perception and Performance35(6), 1811.

Stephen, D. G., & Dixon, J. A. (2008). The self-organization of insight: Entropy and power laws in problem solving. The Journal of Problem Solving2(1), 6.

And for a review of nonrepresentational approaches to perception and action that do not assume modular systems I recommend:

Chemero, A. (2013). Radical embodied cognitive science. Review of General Psychology17(2), 145-150.

Authors’ Response: We are deeply grateful to the reviewer for their insightful critic, which we believe has helped us in improving the manuscript.

Thank you for your references. We have enhanced our Introduction by including description as well as relevance of nonrepresentational approaches to perception and action.

As AMI is an indicator of non-linear autocorrelation, we have not compared with mutual information in our theoretical study. In our study, the use of the formula for mutual information helps us explain increased joint probability of activity of neurons in two circuits in a network. This has allowed us to come to conclusion that processing of the same stimulus by many circuits increases the mutual information, which in turn is responsible for coupling the action with perception. In order to emphasize the differences with other variables, as mentioned in your comment, we have added a new section titled: Self-organization of cognitive structures in action-perception (lines 559 – 597). The above references, which are very relevant to this work, have been cited.

In addition, we added new sections (Affordance and mutual information (line 520), Action-perception involves combined increase of mutual information and Shannon information (line 539) to show the relevance of the current work to embedded cognitive theories. We have updated ‘Abstract’ (added new text in lines 22-26). We have also updated the ‘Summary’ to reflect the changes in the Introduction: (590 – 621).

We have added a new Figure 5 in our attempt to explain the principles of ecological psychology within the information theoretic framework proposed by us.

In ‘Introduction’ we have specifically attempted to explain why this work is relevant (please see lines from 71..).

Other Comments of Reviewer and Authors' response are below:

There are also doubts whether the distinction between dorsal and ventral streams is useful at all for understanding certain perceptual tasks:

Durgin, F. H., Hajnal, A., Li, Z., Tonge, N., & Stigliani, A. (2010). Palm boards are not action measures: An alternative to the two-systems theory of geographical slant perception. Acta Psychologica134(2), 182-197.

Even though I welcome and appreciate the authors stating that the dorsal and ventral systems have quite a bit of overlap, I still think it would be useful to include some of these references as contrasting views so that the reader has an appreciation that not all statements mentioned in the manuscript are settled facts, rather they are still subject to much empirical testing and theoretical debate.

Authors' response:

Thank you for this reference. We have cited this work (lines 428-430).

Reviewer 2 Report

In this paper is proposed a perception model  in wich increasing in mutual 
 information resulting from decreasing of the total entropy in the  neural net is partly responsible for perceptual experience.

The aim of the proposed research is not clear; authors should highlight in the introduction section the state of art and the objectives of their research.

Section 3 should be more structured. The hierarchy of the sub-paragraphs is not  complete (for example, not exists the subparagraph 3.1, but there are the subparagraphs 3.1.1 and 3.1.2, etc.).  Furthermore, the sequence and the connection between the subparagraphs is not clear.

English is poor and needs to be improved

Author Response

Thank you very much for critic:

We have attempted to improve the manuscript as summarized below:

In ‘Introduction’ we have specifically attempted to make the aim of this work is clear (please see lines from 71 - 72).

We have also discussed other similar works at the end of the manuscript (lines 623 - 630).

To reflect changes in the Introduction, we added new sections

(Affordance and mutual information (line 520), Action-perception involves combined increase of mutual information and Shannon information (line 539) to show the relevance of the current work to embedded cognitive theories. We have updated ‘Abstract’ (added new text in lines 22-26). We have also updated the ‘Summary’ to reflect the changes in the Introduction: (590 – 630).

We have added a new Figure 5 in our attempt to explain the principles of ecological psychology within the information theoretic framework proposed by us.

We have also attempted to improve the language, but we will be happy to send the manuscript for professional editing.

'Section 3' is 'Section 4' now, which we have restructured according to your recommendations.

Reviewer 3 Report

Congratulations to the authors of the article. The article is very original and is intended to a very topical area. The competence and qualification of the authors is doubtless. The most recent sources (2016-2018 period) are cited. The paper analyzes very actual theme. The paper contains new and significant information adequate to justify publication. The paper provides interesting results obtained on the basis of the adopted appropriate research methods. The empirical part remains the strength of the paper. The methods are employed appropriate. The results are presented clearly and analyzed appropriately. The paper is written clearly and understandable.

I have just one minor comment. I think it is necessary to cite the original sources of „Shannon“.

Shannon, Claude E. - An Algebra for Theoretical Genetics. MIT Ph.D. thesis, Department of Mathematics, 1940. MIT Institute Archives. - ìAnalogue of the Vernam System for Continuous Time Series.î Memorandum Bell Laboratories, 1943. reprinted in Claude Elwood Shannon: Collected Papers. New York: IEEE Press, 1993. - ìCommunication Theory of Secrecy Systemsî. Bell System Technical Journal, July 1948, p. 379; Oct. 1948, p623. Reprinted in Claude Elwood Shannon: Collected Papers. New York: IEEE Press, 1993. - "Creative Thinking". Claude Elwood Shannon: Miscellaneous Writings. Mathematical Sciences Research Center, ATT. 1993. - "A Mathematical Theory of Communication". Bell System Technical Journal, July and October 1948. Reprinted in Claude Elwood Shannon: Collected Papers. New York: IEEE Press, 1993. - A Symbolic Analysis of Relay and Switching Circuits. MIT Department of Electrical Engineering M.S. 1938. MIT Institute Archives. Shannon, Claude E. and Weaver, Warren. The Mathematical Theory of Communication. Urbana: The University of Illinois Press. 1949.

Author Response

We are grateful for your encouraging comments about our work. According to your recommendations, we have attempted to improve the Introduction as summarized below:

In ‘Introduction’ we have specifically attempted to make the aim of this work is clear (please see lines from 71 - 72). We have also discussed other similar works at the end of the manuscript (lines 623 - 630).

To reflect changes in the Introduction, we added new sections (Affordance and mutual information (line 520), Action-perception involves combined increase of mutual information and Shannon information (line 539) to show the relevance of the current work to embedded cognitive theories. We have updated ‘Abstract’ (added new text in lines 22-26). We have also updated the ‘Summary’ to reflect the changes in the Introduction: (590 – 630).

We have added a new Figure 5 in our attempt to explain the principles of ecological psychology within the information theoretic framework proposed by us.

 We have also added the following original work by Shannon in our reference (line 738):

Shannon, C.E.; Weaver, W. The mathematical theory of communication; University of Illinois Press: Urbana,, 1949; pp. v (i.e. vii), 117 p.

Round 2

Reviewer 2 Report

In this new version of their paper the authors take into account all my suggestions. I consider this manuscript publishable in the present form.